# Antiviral Activity of Selected Essential Oils against Cucumber Mosaic Virus

**DOI:** 10.3390/plants12010018

**Published:** 2022-12-20

**Authors:** Elżbieta Paduch-Cichal, Ewa Mirzwa-Mróz, Patrycja Wojciechowska, Katarzyna Bączek, Olga Kosakowska, Zenon Węglarz, Marek Stefan Szyndel

**Affiliations:** 1Section of Plant Pathology, Department of Plant Protection, Institute of Horticultural Sciences, Warsaw University of Life Sciences (WULS-SGGW), ul. Nowoursynowska, 166, 02-787 Warsaw, Poland; 2Department of Vegetable and Medicinal Plants, Institute of Horticultural Sciences, Warsaw University of Life Sciences (WULS-SGGW), ul. Nowoursynowska 166, 02-787 Warsaw, Poland

**Keywords:** *Cucumber mosaic virus*, *in vitro*, *in vivo*, essential oils, thyme, oregano, costmary

## Abstract

The aim of the study was to assess the antiviral activity of selected essential oils (EOs) against *Cucumber mosaic virus* (CMV), both in vitro and in vivo. The observations were made using *Chenopodium quinoa* as a local host. The EOs were obtained from Greek oregano, thyme, and costmary. Their chemical composition was determined using GC/FID followed by GC/MS. The dominant compound in oregano EO was carvacrol (59.41%), in thyme EO—thymol (59.34%), and in costmary EO—β-thujone (90.60%). Among the analysed EOs, thyme EO exhibited the most promising effects against CMV. However, its activity was influenced by the time of application. In an in vivo experiment, thyme EO showed protective (pre-inoculation) rather than curative (post-inoculation) activity.

## 1. Introduction

Viral plant diseases can be found worldwide. Plant viruses infect many crops, causing serious losses in their production. Protection of crops against viruses is difficult. In general, two methods of crop protection against viruses are employed in plant cultivation. The first one, as applied especially in the case of vegetatively propagated plants, consists of ensuring that a sanitarily certified propagation material is used, the second one being chemical control of insect vectors. Another method involves the use of resistant varieties developed by breeding. However, effective crop protection with this method is possible only for a limited number of viruses. Additionally, it is possible to eliminate viruses in vitro, which is accomplished with different techniques (e.g., thermotherapy or meristem culture) which are all expensive, time-consuming, and are not suitable for all plant/virus pathosystems [1,2]. Thus, the search for alternative methods of crop protection against plant viruses is of great significance.

Essential oils (EOs), due to their wide range of biological activities, have been studied extensively. Since the Middle Ages, they have seen widespread use in bactericidal, virucidal, fungicidal, antiparasitical, and medicinal applications. At present, they are especially common in pharmaceutical, sanitary, cosmetic, agricultural, and food industries. Moreover, EOs are of great importance in several fields, including plant protection against diseases [3,4,5,6,7]. Although limited, current knowledge about the antiviral effects of EOs indicates their potential to control the spread of viral infections [6,8]. A number of recent reports have provided data on the activity of EOs against plant viruses [4,9,10,11,12,13,14,15,16,17,18,19]. However, this field of study is still insufficiently explored and further research is required to enable a more complete understanding of the mechanisms behind the antiviral activities of EOs [1].

The detailed mechanism of such activity is yet to be fully explored. It has been hypothesized that EO components could either directly inactivate viral particles or induce resistance/tolerance response in the host [2]. The mechanisms of action of EOs are versatile due to the complexity of composition and synergism of bioactive compounds compared with synthetic antiviral substances (e.g., acyclovir) that act in a manner specific to a particular type of virus. The action of EOs can inhibit the virus at the extracellular level, i.e., inhibit its penetration into the host cell, by interfering with the structure of the viral envelope or blocking viral proteins that are necessary for the virus to enter the host cells. EOs can also have antiviral effects against intracellular viruses. The mechanisms are not yet fully understood. It is possible to determine the stage of the infection at which the applied EOs work, but there are no precise data on the molecular mechanism of action, including specific sites of action or the types of interactions. It is important to learn about these mechanisms in order to be able to use the potential of EOs in viral infections [3,20,21].

*Cucumber mosaic virus* (CMV) is one of the most common plant pathogenic viruses belonging to the family *Bromoviridae*, genus *Cucumovirus*. It is a positive-sense ssRNA virus with a tripartite genome encapsidated in isometric particles, measuring approximately 30 nm in diameter. It is widespread and common in temperate regions. It is an important virus, which affects more than 1200 different species of 40 dicotyledonous and monocotyledonous families and causes significant economic losses in many vegetable and ornamental crops. It is transmitted non-persistently by more than 60 aphid species. In addition to plants of the Cucurbitaceae family (cucumber, pumpkin, and melon), serious CMV infections have also occurred, for example, in spinach, tomato, carrots, parsley, celery, beetroot, pea, broad been, lettuce, tobacco, chrysanthemum, tulip, geranium, gladiolus, lily, nasturtium, and phlox [22]. In general, two methods of crop protection against CMV are employed in plant cultivation. The first one, as applied especially in the case of vegetatively propagated plants, consists of ensuring that a sanitarily certified propagation material is used, the second one, in protecting host plants, especially cucumber plants grown under cover, from aphids. Plant cultivation in the proximity of CMV host plants should be avoided. Diseased plants with a considerably reduced yield should be removed from the plantation [1,2]. Recently, essential oils extracted from different plants and their constituents have been studied for their antiviral properties against CMV in Europe, Asia, Africa, and South America. The conducted studies have confirmed the activity of the extracted essential oils against CMV [4,12,16,19,23,24,25,26,27,28,29].

This work was undertaken to test the effectiveness of Greek oregano, thyme, and costmary EOs against CMV as antiphytoviral agents. The aim of the study was to provide answers to the following questions: (i) do the EOs exhibit antiphytoviral activity, (ii) is the antiviral activity affected by differences in the EO composition, and (iii) does the application of EOs treatment before or after plant inoculation influence the antiviral activity in local host plants.

## 2. Results

### 2.1. The Composition of EOs

In the case of Greek oregano EO, 28 compounds were identified, comprising 98.98% of the sample. Monoterpenes created the fundamental part of this EO. Here, phenolic monoterpenes—with carvacrol as a clear dominant (59.41%), followed by a small share of thymol (1.19%)—were present in the highest amount. Monoterpene hydrocarbons, which formed 32.13%, were represented mainly by γ-terpinene (19.73%) and *p*-cymene (4.29%), while oxygenated monoterpenes (4.51%) were formed by, i.a., bornyl acetate, terpinen-4-ol, and β-terpineol. Fraction of sesquiterpenes included hydrocarbons (β-caryophyllene, germacrene D) and oxygenated substances, such as (−)-spathulenol and α-bisabolol.

In thyme EO, 26 compounds were detected, accounting for 97.68% of the total identified fraction. Phenolic monoterpenes were the dominant group in the EO (61.75%). Within this fraction, thymol formed 59.34%, while carvacrol 2.41%. Monoterpene hydrocarbons, with a domination of γ-terpinene (12.78%) and *p*-cymene (9.98%), comprised 29.52% of the sample. Oxygenated monoterpenes (3.63%) were represented mainly by linalool and bornyl acetate. The share of sesquiterpenes was at a level of 2.35% (sesquiterpenes hydrocarbons) and 0.17% (oxygenated sesquiterpenes).

In costmary EO, 24 compounds were recognized, comprising 99.35% of the sample. Oxygenated monoterpenes were present in the highest amount (95.60%), with a clear domination of thujones (β-thujone 90.60%; α-thujone 2.55%). The share of other fractions was low, as follows: monoterpenes hydrocarbons 2.62%, phenolic monoterpenes 0.36%, and sesquiterpenes 0.44% (Table 1).

### 2.2. Evaluation of the Antiviral Activity of the EOs against the CMV-S21 Isolate (In Vitro)

Greek oregano, thyme, and costmary EOs were applied on the *Ch. quinoa* host plants simultaneously with the application of the virus, which reduced the number of local lesions on the CMV-infected plants. Significant differences were observed in the activity of thyme EO at the concentrations of 2000 ppm, 3000 ppm, 4000 ppm, 5000 ppm, and 6000 ppm against the CMV-S21 isolate compared with the effect of Greek oregano and costmary EOs. Thyme EO was the most effective in reducing the number of local lesions on the *Ch. quinoa* leaves and showed inhibition of local lesions at the level 30.1–79.5%. The strongest activity of thyme EO was recorded at the concentration of 6000 ppm, and the mean number of spots on four inoculated leaves was 30.4. (inhibition of local lesions was 79.5%). No significant differences were observed between the activity of thyme EO (1000 ppm), Greek oregano EO (3000–6000 ppm), or costmary EO (6000 ppm) against the CMV-S21 isolate. The number of local spots visible on plant leaves after inoculation with the CMV-S21 isolate (control group 3) was not significantly different from the number of spots on the *Ch. quinoa* leaves after inoculation of plants with the CMV-S21 isolate with the addition of costmary EO (500–5000 ppm), Greek oregano EO (500–3000 ppm), or thyme EO (500 ppm) (Table 2).

The average absorbance (OD_405 nm_) of samples from the experimental group was lower than that of the samples from control group 3. The average absorbance (OD_405 nm_) of all the EOs tested against CMV-S21 isolate increased with the increase in their concentrations. The average absorbance (OD_405 nm_) for samples of plants infected with the CMV-S21 isolate with the addition of thyme EO (500–6000 ppm) was lower compared with the average absorbance for Greek oregano and costmary EOs. The average absorbance (OD_405 nm_) was the lowest for samples of plants infected with the CMV-S21 isolate with the addition of thyme EO at the concentration of 6000 ppm (Table 3).

Control group 1—uninoculated *Ch. quinoa* plants, λ = 405 nm—0.122; control group 2—the plants inoculated with only the EOs (500 ppm, 1000 ppm, 2000 ppm, 3000 ppm, 4000 ppm, 5000 ppm, and 6000 ppm), λ = 405 nm—0.135; control group 3—(plants inoculated with only the S21 CMV isolate), λ = 405 nm—1.322.

The regression analysis performed to study the relationship between the thyme EO concentration and the average number of spots present on four *Ch. quinoa* leaves after inoculation with the CMV-S21 isolate showed a strong correlation (Pearson coefficient r = −0.86) between the aforementioned variables (Figure 1). The equation of the fitted model showed a statistically significant relationship between the average number of spots and the thyme EO concentration (Figure 1). The equation of the fitted model is:Average spot number=136.755−0.0191528∗thyme EO concentration.

### 2.3. Evaluation of the Antiviral Activity of the EOs against the CMV-S21 Isolate (In Vivo)

Thyme EO showing the greatest percent of inhibition of local symptoms on the *Ch. quinoa* plants in the in vitro experiment were chosen for in vivo testing.

The samples of *Ch. quinoa* plants from Test A and Test B, control groups I and II were analysed by DAS-ELISA test. The DAS-ELISA test results are presented in Table 4.

The average absorbance (OD_405 nm_) of samples from the experimental group was lower than that of the samples from control group II. The average absorbance (λ = 405 nm) readings for the samples from the plants treated with thyme oil 24, 48, and 72 h before the inoculation with CMV were lower compared with those for the samples from the plants on which the EO was applied 24, 48, and 78 h after the inoculation with CMV (Table 4).

The application of thyme EO (6000 ppm) on *Ch. quinoa* leaves before the inoculation with CMV proved more effective than the application of the EO after the inoculation with CMV. These differences were statistically significant. No significant differences were observed in the number of local lesions visible on the leaves of *Ch. quinoa* plants treated with thyme EO 24, 48, and 72 h before the inoculation with CMV. The number of local lesions was significantly lower when application of thyme EO was performed 72 h after the inoculation with CMV compared with the application of the EO after 24 h or 48 h. The effect of thyme EO applied on the leaves 24 h and 48 h after the inoculation was not statistically different (Figure 2).

## 3. Discussion

Chemically, essential oils are multi-component mixtures of monoterpenes, sesquiterpenes, and their derivatives, including aromatic derivatives. These substances can be in the form of alcohols, ketones, aldehydes, esters, or ethers. Usually, a single essential oil contains several dozens of compounds with different concentrations and activities. Large variation in the chemical composition of EOs determines a versatile range of their applications. Chemical polymorphism among aromatic plants is a widely described phenomenon [31] The number of individual compounds and their proportion in a particular EO is variable and depends on many factors, including genetic variations of the species, the age of the plant, the type of raw material obtained, geographical location of cultivation, the time (season) and conditions of harvest, the environmental conditions of growth, post-harvest treatment of raw materials, and the method of isolation. In the vast majority of EOs, however, it is possible to distinguish one to three dominant compounds responsible for a specific aroma and biological activity of EOs [21,24,32,33,34,35,36].

The aim of the first experiment in the present study was to verify whether Greek oregano, thyme, and costmary EOs exhibit antiphytoviral activity against CMV.

A common feature of Greek oregano and thyme EOs is a high content of monoterpenes, including phenolic monoterpenes, oxygenated monoterpenes, and monoterpene hydrocarbons. However, there were significant differences in the composition of the two EOs. Greek oregano EO had a higher proportion of carvacrol (59.41%) and γ-terpinene (19.73%), while the content of thymol amounted to 1.19%. Conversely, in thyme EO, thymol (59.34%) was the dominant compound, with the content of carvacrol and γ-terpinene amounting to 2.41% and 12.78%, respectively. Moreover, Greek oregano EO contained a higher percentage of oxygenated monoterpenes, whereas thyme EO contained a higher percentage of sesquiterpenes. In costmary EO, the dominant compound was β-thujone (90.60%), classified as an oxygenated monoterpene. The content of monoterpene hydrocarbons, phenolic monoterpenes, and sesquiterpenes was low in this EO compared with the content in Greek oregano and thyme EOs.

Apart from the dominant compounds, the Greek oregano EO contained a higher percentage of oxygenated monoterpenes, whereas the thyme EO contained a higher percentage of sesquiterpenes. In the costmary EO, the oxygenated monoterpenes were the dominant component, while the content of monoterpene hydrocarbons, phenolic monoterpenes, and sesquiterpenes was low compared with the content of these compounds in the Greek oregano and thyme EOs. Both phenolic monoterpenes and monoterpene hydrocarbons were present in the Greek and thyme EOs in high percentages, while the costmary EO had a high content of oxygenated monoterpenes, which indicates a relationship between the class of chemical compounds in which it was present in a particular EO in the largest proportion and how effective was the antiviral activity it exhibited. In addition to the main component of the tested EO obtained from a specific plant species, its biological properties and, thus, its antiviral activity, will also be affected by the percentage content of the individual components [28].

The experiments carried out in the present study confirmed that the following EOs extracted from different plants had antiphytoviral activity against the following viruses: *Foeniculum vulgare* EO and *Pimpinella anisum* EO—against *Potato virus X* (PVX) [9]; *Plectranthus tenuiflorus* EO—against *Tobacco necrosis virus* (TNV); *Tomato spotted wilt virus* (TSWV) and *Tobacco mosaic virus* (TMV) [10], *Foeniculum vulgare* EO, and *Pimpinella anisum* EO—against TMV and *Tobacco ringspot virus* (TRSV); *Picrasma quassioides* EO, *Melaleuca leucadendron* EO, *Myrtus communis* EO, and *Satureja montana* EO—against TMV [11,13,14]; *Azadirachta indica* EO, *Clerodendrum inerme* EO, *Schinus terebinthifolius* EO, and *Mirabilis jalapa* EO—against *Bean common mosaic* (BCMV) [15]; *Tanacetum vulgare* EO—against *Potato virus Y* (PVY) from [16]; *Lavandula angustifolia* EO and *Foeniculum officinale* All. var. *dulce* EO—against TSWV [17]; and *Thuja orientalis* EO, *Nigell sativa* EO, *Azadirachta indica* EO, and *Bougainvillea spectabilis* EO—against *Zucchcini yellow mosaic virus* (ZYMV) [18].

Despite a wide spectrum of biological activities of EOs, only a limited amount of information is available about their effect on CMV.

A number of similarities was observed between the composition and activity of EOs investigated in this study and the currently available literature data. According to Bezić et al. [4], the application of *Satureja montana* EO on *Ch. quinoa* and *Ch. amaranticolor* host plants simultaneously with the inoculation with CMV reduced the number of local lesions by 24.1%. The main components of *S. montana* EO were thymol and carvacrol. Thymol was more effective in reducing the percentage of CMV infection (reduction by 33.2%), while the percentage reduction of CMV infection by carvacrol amounted to 28.3% [4]. Moreover, the EOs of several *Teucrium* species (*T. polium*, *T. flavum*, *T. montanum*, *T. chamaedrys*, and *T. arduini*), rich in monoterpenes and sesquiterpenes, significantly reduced the percentage of CMV infection of *Ch. quinoa* plants (reduction by 22.9–43.4%) [4,24]. Sesquiterpenes and monoterpenes were also present in relatively high percentages in *Micromeria fructiculosa* and *M. graeca* EOs, thus reducing the number of local lesions on *Ch. quinoa* inoculated with CMV by 23.8–43.6% [26,28,29]. Additionally, EOs of both *Eryngium alpinum* and *E. amethystinum* significantly reduced the number of local lesions on *Ch. quinoa* inoculated with CMV (reduction by 77.8% and 80.5%, respectively). The EOs of both species contained high percentages of oxygenated sesquiterpene compounds [23]. The experiments revealed that the effectiveness of thyme essential oil against CMV depends on its concentration. Local lesion inhibition of 50% was achieved by applying the thyme EO at a concentration of 3000 ppm, whilst the inhibition of 80% was achieved by using a concentration of 6000 ppm.

The results obtained in this study indicate that the time of application of thyme EO has a significant influence on its activity against the CMV infection. The best inhibitory effect was obtained through protective (prior to virus inoculation) rather than curative (post-inoculation) treatment. According to Helal [19], the maximum protection and inhibition percentage (91.5%) was observed 24 h before the application of the thyme essential oil (3000 ppm) on cucumber plants. These results correspond with those of Vuko et al. [29] who treated the *Ch. quinoa* plants with *Micromeria croatica* essential oil 24, 48, and 72 h prior to the inoculation with CMV. Although all pre-treatments significantly reduced the number of local lesions, the strongest antiviral effect was manifested after the treatment that was performed 72 h prior to the inoculation. The percentage reduction of the number of local lesions ranged between 66.8% and 71.4%. According to Dunkić et al. [23], pre-inoculation treatment with both the *Eryngium alpinum* and *E. amethystinum* EOs significantly reduced the number of local lesions, with the percentage reduction amounting to 77.8 and 80.5%, respectively. The antiviral effect of EOs appears to involve direct inhibition of virus replication, or indirect inhibition through induction of systemic resistance of the host plant against the virus, which may persist for long periods depending on both plant species and the virus strain [19].

All of the studied EO components, some of which are present as major constituents and some of which are present in relatively small amounts, can have a synergetic effect and may contribute to the antiviral efficacy of the EOs. Antiviral testing of many EOs could help us gain insight into the relationship between the EO composition and their antiviral efficiency. The results obtained can be a starting point for further research into the antiphytoviral activity of essential oils and individual components of the oils. Understanding the mode of such activity may help find and adjust these natural substances for possible use in the control of viral plant diseases [4].

To our knowledge, this is the first study to address this topic in Poland.

## 4. Materials and Methods

### 4.1. Plant Raw Materials Used for Distillation of EOs

The EOs used in the study were obtained from herbs of three species of medicinal and aromatic plants, i.e., Greek oregano (*Origanum vulgare* L. subsp. *hirtum* (Link) Ietswaart), thyme (*Thymus vulgaris* L.), and costmary (*Tanacetum balsamita* L.). The plants were cultivated at the experimental field of Department of Vegetable and Medicinal Plants, Warsaw University of Life Sciences—SGGW (5210180 N; 2105234 E). The herb (upper, not wooden parts of shoots) of Greek oregano and thyme was collected from 2-year-old plants, and costmary from 4-year-old plants, at the stage of flowering. The raw materials were dried at the temperature of 35–40 °C in the dark.

### 4.2. EOs Extraction and GC-MS/GC-FID Analysis

Essential oils were extracted according to European Pharmacopoeia, with modifications [37]. Fifty grams of air-dried raw material was used for hydrodistillation for 3 h using a Deryng-type apparatus. Until the analysis, the samples were stored in dark vials at 4 °C.

Analysis of essential oils were carried out by gas chromatography (GC) coupled with mass spectrometry (MS) and flame ionization detector (FID). The qualitative and quantitative analysis was carried out by means of an Agilent Technologies 7890A gas chromatograph equipped with FID and MS Agilent Technologies 5975C Inert XL_MSD with Triple Axis Detector (Agilent Technologies, Wilmington, DE, USA). Details of the operation conditions were given previously by Bączek et al. [38]. Capillary, polar column HP 20M (25 m × 0.32 mm × 0.30 µm) (Agilent Technologies, Wilmington, DE, USA) was applied. Separation conditions were as follows: oven temperature isotherm at 60 °C for 2 min, temperature rising at a rate of 4 °C per min, from 60 °C to 220 °C, then held isothermal at 220 °C for 5 min. The carrier gas (He) flow was 1.1 mL/min. The split ratio was 1:50. Diluted samples (1/100 *v*/*v*, in n-hexane:isopropanol) of 1 μL were injected at 210 °C by auto sampler. Ion source temperature was −220 °C, ionization voltage was 70 eV, and the range of mass spectra scanning was 40–500 amu. EOs compound identification was based on comparison of mass spectra from the Databases (NIST08, NIST27, NIST147, NIST11, Wiley7N2) and on comparison of retention indices (RI) relative to retention times of a series of n-alkanes (C_7_–C_30_) (Merck KGaA, Darmstadt, Germany) with those reported in the literature [38]. The percentage share of compounds identified in the EOs was computed by the normalization method from the GC peak areas.

The air-dried herb was hydrodistillated for 3 h using a Deryng-type apparatus. Until the analysis, the samples were stored in dark vials at 4 °C.

Analysis of essential oils was conducted by GC-MS and GC-FID (gas chromatography coupled with mass spectrometry and flame ionization detector).

The qualitative and quantitative analysis was carried out by means of an Agilent Technologies 7890A gas chromatograph equipped with a flame ionization detector (FID) and MS Agilent Technologies 5975C Inert XL_MSD with Triple Axis Detector (Agilent Technologies, Wilmington, DE, USA). Capillary, polar column HP 20M (25 m × 0.32 mm × 0.30 µm) (Agilent Technologies, Wilmington, DE, USA) was applied. Separation conditions were as follows: oven temperature isotherm at 60 °C for 2 min, temperature rising at a rate of 4 °C per min, from 60 °C to 220 °C, then held isothermal at 220 °C for 5 min. The carrier gas (He) flow was 1.1 mL/min. The split ratio was 1:50. Diluted samples (1/100 *v*/*v*, in n-hexane:isopropanol) of 1 μL were injected at 210 °C by auto sampler. Ion source temperature was −220 °C, ionization voltage was 70 eV, and the range of mass spectra scanning was 40–500 amu. EOs compound identification was based on comparison of mass spectra from the Databases (NIST08, NIST27, NIST147, NIST11, Wiley7N2) and on comparison of retention indices (RI) relative to retention times of a series of n-hydrocarbons (C_7_–C_30_) with those reported in the literature. The percentage share of compounds identified in the EOs was computed by the normalization method from the GC peak areas.

### 4.3. Virus and Plant Host

*Cucumber mosaic virus* (CMV) (*Bromoviridae* family, genus *Cucumovirus*), S21 CMV isolate, as well as seeds of the *Chenopodium quinoa* Willd. host plants were provided by The Department of Virology and Bacteriology, Institute of Plant Protection, National Research Institute, Poznań.

The seeds of *Ch. quinoa* were sown in trays in a greenhouse at a temperature of 24 °C under a 16 h/8 h light/dark cycle, with watering as required. When the seedlings were large enough to handle, they were potted individually into 15 cm plastic pots containing fresh compost. The host plants were grown in a greenhouse under the same conditions. The experimental plants were selected for inoculation four weeks after sowing when they had eight true leaves. Care was taken to ensure that the experimental plants were as uniform in size as possible.

The viral inoculum was prepared from cucumber leaves infected with the S21 CMV isolate. The plant material was ground with cold inoculation buffer (0.05 M phosphate buffer, pH 7.0) in a cold mortar. The inoculum prepared was used for mechanical inoculation of *Ch. quinoa* host plant.

### 4.4. Antiviral Effects of the EOs on the Ch. Quinoa Plants

Greek oregano, thyme, and costmary EOs were studied both in vitro and in vivo.

In vitro antiviral activity—Equal volumes of solutions containing 500 ppm, 1000 ppm, 2000 ppm, 3000 ppm, 4000 ppm, 5000 ppm, and 6000 ppm of the EOs in Tween 80 and distilled water were added separately to the virus inoculum (20 μg/mL concentration) and immediately used for mechanical inoculation of the *Ch. quinoa* plants (the experimental group).

Control group 1—uninoculated *Ch. quinoa* plants, Control group 2—plants inoculated with only the EOs (500 ppm, 1000 ppm, 2000 ppm, 3000 ppm, 4000 ppm, 5000 ppm, and 6000 ppm), and Control group 3—plants inoculated with only the S21 CMV isolate.

Twenty *Ch. quinoa* plants (four leaves per plant) were inoculated in this experiment. Local lesions that developed on the leaves 5 days after virus inoculation were counted. All the treated plants were grown in a greenhouse (24 °C, 16 h/8 h light/dark cycle).

The inhibition percentage was calculated according to the formula
IP [%]=[(C−T)/C]×100
where IP—inhibition of local lesions (%), C—average number of local viral lesions in control group 3, and T—average number of local viral lesions in control group 2.

The EOs showing the greatest percent of inhibition of local symptoms on the *Ch. quinoa* plants in the in vitro experiment were further tested in vivo.

The protective and curative effect of the evaluated EOs against CMV infection was investigated in vivo according to the method described by Helal (2019) with some modifications: 

Experimental group

Curative effect—Test A—mechanical inoculation of the S21 CMV isolate (50 μL/leaf) was performed, then 24 h, 48 h, and 72 h after virus inoculation, the virus-inoculated leaves were treated with a 6000 ppm solution of thyme EO (50 μL/leaf).

Protective effect—Test B—50 μL/leaf of the 6000 ppm solution of the thyme EO were rubbed on *Ch. quinoa* leaves. Subsequently, after 24 h, 48 h, and 72 h, mechanical inoculation of these leaves was performed.

Control group I—untreated *Ch. quinoa* plants.

Control group II—plants inoculated mechanically with the S21 CMV isolate.

Twenty *Ch. quinoa* plants (four leaves per plant) were inoculated in this experiment. Local lesions that developed on the leaves 5 days after virus inoculation were counted. All the treated plants grown in a greenhouse (24 °C, 16 h/8 h light/dark cycle).

### 4.5. DAS-ELISA Test

DAS-ELISA test [39] was used to estimate the antiviral activity of the EOs against the CMV-21 isolate using a specific antibody from LOEWE Biochemica GmbH (Germany). The plant samples (samples from the experimental group, control groups 1, 2, and 3, Test A and Test B, control groups I and II) were prepared by grinding 0.250 g of fresh plant tissue in an extract buffer in the ratio of 1:10 (*w*/*v*) and tested according to the manufacturer’s protocol. After 1 h of incubation at room temperature, substrate hydrolysis was measured as a change in absorbance at OD_405 nm_ using the Infinite^®^ 200Pro microplate reader (Tecan GmbH, Austria). Samples were considered positive if their optical density (OD_405 nm_) readings were at least twice those of the healthy controls. The average absorbance values of the experimental group and control groups 1, 2, and 3 are presented.

### 4.6. Statistical Analysis

One-way analysis of variance (ANOVA) was performed to investigate the effects of the EOs used at the concentrations of 500, 1000, 2000, 3000, 4000, 5000, and 6000 ppm against the CMV activity. The experiment was conducted in 20 replications. Each replication was one virus-inoculated *Ch. quinoa* plant treated with one of the 3 tested EOs at the above concentrations, for which the number of local spots present on 4 leaves was determined by counting. The control variant were 20 CMV-inoculated plants. An additional control variant were plants treated with only the 3 tested EOs of different concentrations, and the Student–Newman–Keuls multiple comparison test was used at a significance level of *p* = 0.05.

To test the relationship between the concentration of an EO and the average number of spots present on 4 leaves of *Ch. quinoa* after inoculation with the CMV-S21 isolate, regression analysis was performed. The analysis was carried out for the most effective EO (showing the greatest percentage of inhibition of local symptoms on the *Ch. quinoa* plants).

A linear regression equation was calculated according to the model:y=a−bx where y = average spot number, and x = thyme EO concentration.

One-way analysis of variance (ANOVA) was performed to test the effectiveness of the EO showing the greatest percentage of inhibition of local symptoms on the *Ch. quinoa* plants in the in vitro experiment depending on the time of its application (24 h, 48 h, or 72 h before inoculation or 24 h, 48 h, or 72 h after inoculation with the isolate S21 CMV. For comparisons of the effects of thyme EO applied on the leaves of the *Ch. quinoa* plants before and after inoculation, the Student–Newman–Keuls multiple comparison test was used with a significance level of *p* = 0.05. All statistical analyses were performed using Statgraphics Plus for Windows 4.1.

## 5. Conclusions

Among the analyzed EOs, only thyme EO showed significant antiviral activity against CMV. The strongest activity of thyme EO was recorded at the concentration of 6000 ppm and the mean number of spots on 4 inoculated leaves was 30.4. (Inhibition of local lesions 79.5%). However, the activity clearly depended on the method of application and the duration of the application. The application 24 h, 48 h, and 72 h before the virus inoculation gave much better results than the application after the virus inoculation. The presented study is novel and constitutes the first step towards research into future methods of plant protection against viruses.

## Figures and Tables

**Figure 1 plants-12-00018-f001:**
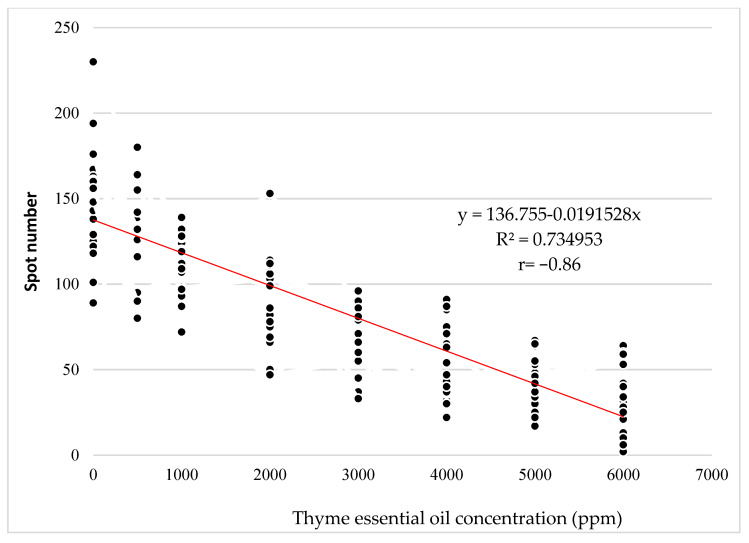
Effect of the thyme EO concentration on the number of local lesions on the *Ch. quinoa* plants inoculated with the CMV-S21 isolate.

**Figure 2 plants-12-00018-f002:**
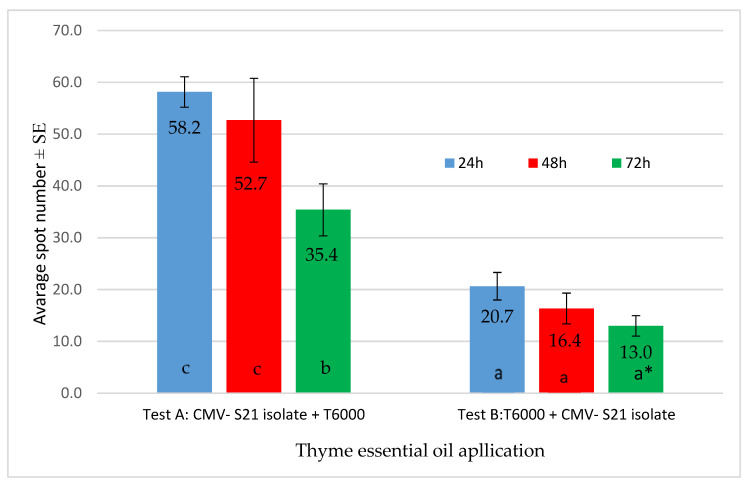
In vivo antiviral effect of thyme EO (6000 ppm) on the infectivity of the CMV-S21 isolate on *Ch. quinoa* plants. * homogenous group according to the Student–Newman–Keuls test. Values marked with the same letter in a bar do not differ statistically at the significance level *p* = 0.05.

**Table 1 plants-12-00018-t001:** The composition of the EO samples.

				Relative Area (%)		
No.	Compound	RI ^1^	RI ^2^	Greek Oregano	Thyme	Costmary
1	(Z)-salvene	935	-	-	-	0.28 ± 0.02
2	(E)-salvene	949	-	-	-	0.05 ± 0.01
3	α-thujene	1023	1012–1039	0.21 ± 0.02	0.33 ± 0.03	0.01 ± 0.00
4	α-pinene	1029	1008–1039	2.26 ± 0.19	2.03 ± 0.16	0.08 ± 0.01
5	camphene	1075	1043–1086	0.40 ± 0.04	0.37 ± 0.05	0.19 ± 0.06
6	β-pinene	1111	1085–1138	0.27 ± 0.00	0.18 ± 0.01	0.10 ± 0.01
7	sabinene	1125	1098–1140	0.08 ± 0.01	0.08 ± 0.02	0.02 ± 0.00
8	β-myrcene	1167	1155–1169	2.37 ± 0.24	1.83 ± 0.22	-
9	α-terpinene	1183	1154–1195	2.17 ± 0.22	1.57 ± 0.06	0.39 ± 0.09
10	D-limonene	1203	1178–1219	0.20 ± 0.05	0.21 ± 0.04	0.36 ± 0.04
11	1.8-cineole	1216	1186–1231	-	0.08 ± 0.01	1.82 ± 0.16
12	β-ocimene	1236	1211–1251	0.06 ± 0.00	0.07 ± 0.01	-
13	γ-terpinene	1253	1222–1266	19.73 ± 1.88	12.78 ± 1.28	0.37 ± 0.10
14	*p*-cymene	1276	1246–1291	4.29 ± 0.30	9.98 ± 0.60	0.88 ± 0.10
15	terpinolene	1285	1261–1300	0.09 ± 0.01	0.09 ± 0.01	0.22 ± 0.02
16	hexanol	1349	1344–1360	-	0.12 ± 0.01	-
17	octan-3-ol	1392	1372–1408	-	0.14 ± 0.03	-
18	α-thujone	1418	1385–1441	-	-	2.55 ± 0.33
19	β-thujone	1437	1400–1452	-	-	90.60 ± 5.03
20	1-octen-3-ol	1446	1411–1465	0.51 ± 0.06	-	-
21	menthone	1459	1450–1475	0.29 ± 0.03	-	0.03 ± 0.00
22	α-copaene	1494	1462–1522	-	0.49 ± 0.12	-
23	camphor	1509	1481–1537	0.22 ± 0.03	0.14 ± 0.02	0.13 ± 0.01
24	β-cubebene	1536	1518–1560	-	1.10 ± 0.12	-
25	linalool	1542	1507–1564	0.25 ± 0.03	1.39 ± 0.12	0.06 ± 0.01
26	bornyl acetate	1577	1549–1597	1.20 ± 0.18	1.06 ± 0.17	-
27	β-caryophyllene	1593	1570–1685	0.40 ± 0.14	0.64 ± 0.09	0.33 ± 0.03
28	terpinen-4-ol	1597	1564–1630	0.84 ± 0.08	0.67 ± 0.06	-
29	pulegone	1623	1626–1663	-	-	0.26 ± 0.04
30	menthol	1631	1599–1651	0.21 ± 0.01	-	0.15 ± 0.02
31	β-terpineol	1636	1616–1644	0.88 ± 0.06		
32	borneol	1687	1653–1728	-	0.29 ± 0.03	-
33	carvone	1711	1699–1751	0.62 ± 0.09	-	-
34	germacrene D	1716	1676–1726	0.44 ± 0.07	0.12 ± 0.02	-
35	caryophyllene oxide	1976	1936–2023	0.17± 0.02	-	-
36	(−)-spathulenol	2124	2074–2150	0.18 ± 0.01	-	-
37	thymol	2164	2100–2205	1.19 ± 0.12	59.34 ± 5.23	0.21 ± 0.03
38	α-bisabolol	2197	2178–2234	0.21 ± 0.02	-	-
39	carvacrol	2211	2140–2246	59.41 ± 5.13	2.41 ± 0.36	0.15 ± 0.02
40	β-eudesmol	2235	2196–2272	-	-	0.11 ± 0.01
	Total identified			98.98	97.68	99.35
	Monoterpene hydrocarbons			32.13	29.52	2.62
	Oxygenated monoterpenes			4.51	3.63	95.60
	Phenolic monoterpenes			60.60	61.75	0.36
	Sesquiterpene hydrocarbons			0.84	2.35	0.33
	Oxygenated sesquiterpenes			0.39	0.17	0.11
	Other compounds			0.51	0.26	0.33

RI ^1^—experimental retention index on polar column, RI ^2^—range of retention indices on polar column reported by Babushok et al. [30].

**Table 2 plants-12-00018-t002:** In vitro antiviral effect of the EOs on the infectivity of the CMV-S21 isolate on the *Ch. quinoa* plants.

** Experimental Group	Average Number of Local Viral Lesions		SD	Inhibition of Local Lesions (%)
CMV-S21 + Thyme 6000 ppm	30.4	a *	18.60	79.5
CMV-S21 + Thyme 5000 ppm	42.0	b	15.72	71.6
CMV-S21 + Thyme 4000 ppm	56.0	c	21.66	62.2
CMV-S21 + Thyme 3000 ppm	73.0	d	18.00	50.7
CMV-S21 + Thyme 2000 ppm	90.1	e	27.05	39.1
CMV-S21 + Thyme 1000 ppm	112.6	f	16.61	23.9
CMV-S21 + Greek oregano 6000 ppm	121.0	fg	11.94	18.2
CMV-S21 + Greek oregano 5000 ppm	123.6	fgh	10.84	16.5
CMV-S21 + Greek oregano 4000 ppm	125.6	fghi	12.01	15.1
CMV-S21 + Costmary 6000 ppm	127.6	fghi	10.99	13.8
CMV-S21 + Greek oregano 3000 ppm	128.9	fghij	11.00	12.9
CMV-S21 + Thyme 500 ppm	130.5	ghij	28.47	11.8
CMV-S21 + Costmary 5000 ppm	130.9	ghij	11.71	11.5
CMV-S21 + Costmary 4000 ppm	132.7	ghij	12.08	10.3
CMV-S21 + Greek oregano 2000 ppm	132.8	ghij	13.80	10.3
CMV-S21 + Costmary 3000 ppm	135.7	ghij	13.61	8.3
CMV-S21 + Greek oregano 1000 ppm	137.0	ghij	15.17	7.4
CMV-S21 + Costmary 2000 ppm	138.9	ghij	15.70	6.2
CMV-S21 + Greek oregano 500 ppm	141.1	hij	16.56	4.7
CMV-S21 + Costmary 1000 ppm	142.9	hij	26.69	3.4
CMV-S21 + Costmary 500 ppm	144.2	ij	24.78	2.6
*** Control group 3	148.0	j	32.78	

* homogenous group according to the Student–Newman–Keuls test. Values marked with the same letter in a column do not differ statistically at the significance level *p* = 0.05. ** inoculation with the CMV-S21 isolate + the essential oils (500 ppm, 1000 ppm, 2000 ppm, 3000 ppm, 4000 ppm, 5000 ppm, and 6000 ppm). *** control group 3—(plants inoculated with only the CMVS21 isolate).

**Table 3 plants-12-00018-t003:** Results of DAS-ELISA for the *Ch. quinoa* plants inoculated with the CMV-S21 isolate with the addition of the EOs.

Essential Oil	Concentration (ppm)						
	500	1000	2000	3000	4000	5000	6000
	Absorbance at λ = 405 nm						
Greek oregano	1.000	0.954	0.919	0.893	0.823	0.765	0.715
Thyme	0.989	0.802	0.765	0.690	0.591	0.515	0.491
Costmary	1.012	0.989	0.976	0.912	0.892	0.812	0.801

**Table 4 plants-12-00018-t004:** Results of DAS-ELISA for the *Ch. quinoa* plants inoculated with the CMV-S21 isolate with the addition of thyme EO at different times of treatment.

Plant Inoculation/Thyme Oil Application (6000 ppm)	Time of Application (h)		
	24	48	72
	Absorbance at λ = 405 nm		
* Test A CMV-S21 isolate + T6000	0.452	0.431	0.390
** Test B T6000 + CMV-S21 isolate	0.341	0.322	0.301

* inoculation with the CMV-S21 isolate 24–72 h before the application of thyme EO (concentration 6000 ppm), ** application of thyme EO (6000 ppm) 24–72 h after inoculation with the CMV-S21 isolate, control group I—(uninoculated, *Ch. quinoa* plants), λ = 405 nm—0.112, control group II—(plants inoculated with only the S21 CMV isolate), λ = 405 nm—1.402.

## Data Availability

Not applicable.

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
