# Peer review of "Antiviral Activity of Selected Essential Oils against Cucumber Mosaic Virus"

_plants, 2022, doi:10.3390/plants12010018_

Round 1
Reviewer 1 Report
Dear Authors,
The paper is well-written, describing EO's composition from three species, Origanum vulgare L., Thymus vulgaris L., and Tanacetum balsamita L., harvested from Poland and their antiviral activity on CMV. The manuscript is clearly written and the results are clearly presented. However, I have a few suggestions before acceptance for publication.
Introduction
1. Please arrange the reference in the text according to the journal's requirements (for example: A number of recent reports have provided data about the activity of EOs against plant viruses [9, 10, 11, 12, 4, 13, 14, 15, 16, 17, 18,19]).
2. I suggest you to replace with:
Although limited, current knowledge about the antiviral effects of EOs indicates their potential to control the spread of viral infections [6, 8].
Instead:
Current knowledge about the antiviral effects of EOs, although limited, indicates their potential to control or at least reduce the spread of viral infections [6, 8].
3. I suggest you present more details about antiviral studies in the introduction because there is only a sentence. (Recently, plant essential oils and their constituents have been studied for their antiviral properties against CMV in Europe, Asia, Africa, and South America [12, 23, 24, 25, 26, 4, 16, 27, 28, 19, 29].
Results and discussion
4. Please place the title of figure 1 after it (not before it).
5. In table 1, position 21, please replace menthon with menthone.
6. There is an IUPAC recommendation (https://doi.org/10.1351/pac199668122223) to replace the "optical density" with absorbance. I suggest replacing OD 405 with A (405nm) in the whole manuscript (on page 5, under Table 2 rows 1, 2, 4 and 6; in the legend of Table 3 and 5; on page 7 in the first two paragraphs; on page 11, rows 4, 6).
7. In Table 3 and 5, “Absorbance at OD405 nm” should be replaced with “Absorbance at l=405 nm”.
8. The average spot numbers in figure 2 must be written 58.2; 52.7; 35.4; 20.7; 16.4 and 13.0 (not 58,2; 52,7; 35,4; 20,7; 16,4 and 13,0). Is it possible to add error bars to this figure?
9. Add info about SD for values presented in Table 1, 2. Tables 3 and 4 are missing.
10. I suggest modifying table 1 as follows:
|
No. |
Compounds |
RI1 |
RI2 |
Relative area (%) |
||
|
Greek oregano |
Thyme |
Costmary |
||||
11. The main individual compounds contained in the essential oils should be tested for antiviral activity, or more details about other studies should be included. In tea tree, for example, g-terpinene (17.3%) and p-cymene (7.93%) presence, along with other compounds, could be associated with the antiviral effect, so please cite the paper [Lu, 2013]: http://dx. doi.org/10.4014/jmb.1210.10078
Discussion
12. Please arrange the reference in the text according to the journal's requirements (for example: [33, 24, 35, 21, 36, 37]).
13. Please discuss more the relationship between composition and biological activity (some minor compounds could have antiviral activity).
14. Please discuss your results more compared with others researchers. Please elaborate a little on this part: “Similar results have also been reported also in the research conducted by Shukla et al. [9], Othman and Shoman [10], Chen et al. [11], Dunkić et al. [12], Bezić et al. [24], Negai et al. [25], Vuko et al. [26] Bezić et al. [4], Dunkić et al., [23], Jerković-Mujkić [13], Min et al. [14], Elsharkawy and El-Sawy [15] Petrov et al. [16], Dikova et al. [17], Elbeshehy [18], Hamidson et al. [27], Ruščić et al. [28], Helal [19], Vuko et al. [29]”.
Materials and methods
15. Reference for extraction method and chromatographic analysis
16. Insert a title for chromatographic analysis because this part was put under the subtitle “Distillation of EOs”.
17. Please replace the subtitle “Distillation of EOs” with “EOs Extraction” because you applied a distillation method for EOs separation from the plant, you did not make an EOs distillation.
18. Please replace:
The carrier gas (He) flow was 1.1 ml/min.
With:
The carrier gas (He) flow was 1.1 mL/min.
19. Most probable, it is the n-alkanes series instead n-hydrocarbons (C7-C30). Please indicate the reagents source and write numbers as subscripts for C7-C30.
Conclusions
The last sentence should be:
The presented study is novel and constitutes the first step toward research into future methods of plant protection against viruses.
Instead:
The presented study is novel and constitutes the first step towards research into future methods of plant protection against viroses.
Reference
Please correct the reference numbering (you have double numbering at the manuscript end).
Author Response
Introduction
- Please arrange the reference in the text according to the journal's requirements (for example: A number of recent reports have provided data about the activity of EOs against plant viruses [9, 10, 11, 12, 4, 13, 14, 15, 16, 17, 18, 19]).
Answer. Appropriate corrections have been made.
- I suggest you to replace with:
Although limited, current knowledge about the antiviral effects of EOs indicates their potential to control the spread of viral infections [6, 8].
Instead:
Current knowledge about the antiviral effects of EOs, although limited, indicates their potential to control or at least reduce the spread of viral infections [6, 8].
Answer. Appropriate corrections have been made.
- Suggest you present more details about antiviral studies in the introduction because there is only a sentence. (Recently, plant essential oils and their constituents have been studied for their antiviral properties against CMV in Europe, Asia, Africa, and South America [12, 23, 24, 25, 26, 4, 16, 27, 28, 19, 29].
Answer. Appropriate corrections have been made. More details have been provided in the Discussion.
Results and discussion
- Please place the title of figure 1 after it (not before it).
Answer. Appropriate corrections have been made.
- Comment. In table 1, position 21, please replace menthon with menthone.
Answer. Appropriate corrections have been made.
- There is an IUPAC recommendation (https://doi.org/10.1351/pac199668122223) to replace the "optical density" with absorbance. I suggest replacing OD 405 with A (405nm) in the whole manuscript (on page 5, under Table 2 rows 1, 2, 4 and 6; in the legend of Table 3 and 5; on page 7 in the first two paragraphs; on page 11, rows 4, 6).
Answer. Appropriate corrections have been made.
- In Table 3 and 5, “Absorbance at OD405 nm” should be replaced with “Absorbance at l=405 nm”.
Answer. Appropriate corrections have been made.
- The average spot numbers in figure 2 must be written 58.2; 52.7; 35.4; 20.7; 16.4 and 13.0 (not 58,2; 52,7; 35,4; 20,7; 16,4 and 13,0). Is it possible to add error bars to this figure?
Answer. Appropriate corrections have been made.
- Add info about SD for values presented in Table 1, 2. Tables 3 and 4 are missing.
Standard deviation (SD) values has been added into Table 1, 2.
Answer. Appropriate corrections have been made.
- suggest modifying table 1 as follows:
|
No. |
Compounds |
RI1 |
RI2 |
Relative area (%) |
||
|
Greek oregano |
Thyme |
Costmary |
||||
Answer. We would prefer to leave Table 1 in its present form, since the information on the unit (% peak area) has already been given in the table’s title.
- The main individual compounds contained in the essential oils should be tested for antiviral activity, or more details about other studies should be included. In tea tree, for example, g-terpinene (17.3%) and p-cymene (7.93%) presence, along with other compounds, could be associated with the antiviral effect, so please cite the paper [Lu, 2013]: http://dx. doi.org/10.4014/jmb.1210.10078
Answer. Thank you for suggesting this article. The research presented in our paper is the first, preliminary study concerning the antiviral activity of Greek oregano, thyme and costmary EOs against CMV to have been conducted in Poland. The work suggested is a very good reference and we are going to include it in the more detailed studies that we plan to conduct in the future.
Discussion
- Please arrange the reference in the text according to the journal's requirements (for example: [33, 24, 35, 21, 36, 37]).
Answer. Appropriate corrections have been made.
- Please discuss more the relationship between composition and biological activity (some minor compounds could have antiviral activity).
Answer. Appropriate corrections have been made.
- Please discuss your results more compared with others researchers. Please elaborate a little on this part: “Similar results have also been reported also in the research conducted by Shukla et al. [9], Othman and Shoman [10], Chen et al. [11], Dunkić et al. [12], Bezić et al. [24], Negai et al. [25], Vuko et al. [26] Bezić et al. [4], Dunkić et al., [23], Jerković-Mujkić [13], Min et al. [14], Elsharkawy and El-Sawy [15] Petrov et al. [16], Dikova et al. [17], Elbeshehy [18], Hamidson et al. [27], Ruščić et al. [28], Helal [19], Vuko et al. [29]”.
Answer. Appropriate corrections have been made.
Materials and methods
- Reference for extraction method and chromatographic analysis
Answer. Suggested data have been completed.
- Insert a title for chromatographic analysis because this part was put under the subtitle “Distillation of EOs”.
Answer. Proper corrections have been applied. The new title of the subchapter is: ‘EOs extraction and GC-MS/GC-FID analysis’.
- Please replace the subtitle “Distillation of EOs” with “EOs Extraction” because you applied a distillation method for EOs separation from the plant, you did not make an EOs distillation.
Answer. Appropriate corrections have been made.
- Please replace:
The carrier gas (He) flow was 1.1 ml/min.
With:
The carrier gas (He) flow was 1.1 mL/min.
Answer. Appropriate corrections have been made.
- Most probable, it is the n-alkanes series instead n-hydrocarbons (C7-C30). Please indicate the reagents source and write numbers as subscripts for C7-C30.
Answer. The text was corrected and completed according to the suggestions.
Conclusions
The last sentence should be:
The presented study is novel and constitutes the first step toward research into future methods of plant protection against viruses.
Instead:
The presented study is novel and constitutes the first step towards research into future methods of plant protection against viroses.
Answer. Appropriate corrections have been made.
Reference
Please correct the reference numbering (you have double numbering at the manuscript end).
Answer. Appropriate corrections have been made.
Reviewer 2 Report
This article is devoted to the study of the antiviral activity of certain essential oils against the cucumber mosaic virus. The article is interesting and specific. Cucumber mosaic virus is a common disease of cucumbers. The studies proposed by the authors are an interesting contribution to the development of this topic. I recommend that the authors improve the following points:
1. Abstract can be supplemented.
2. In the introduction, you can add a link to the work: 10.3390/molecules27186129.
3. The authors identify many components in essential oils. This is good. It is necessary to specify in more detail which of these components turn out to be the best antiviral substances? What exactly act on the cucumber mosaic virus? Obviously, different components of essential oils will act on different biochemical functions of different viruses.
4. In this work, it is possible to indicate in more detail the relationship between the obtained data and those given in the literature. This will help you understand some of the principles and draw more solid conclusions.
5. Unification of drawings is required.
6. It is necessary to describe in more detail the method of obtaining essential oils. If necessary, make a link to the methodology.
7. Conclusions look very modest. Please revise the conclusions substantially and expand them. The article presents many interesting results, on the basis of which important conclusions can be drawn.
Author Response
- Abstract can be supplemented.
Answer. We believe that the content of the abstract, i.e. a short summary of the research problem, the methods used and the conclusions, is sufficient.
- In the introduction, you can add a link to the work: 10.3390/molecules27186129.
Answer Thank you for suggesting this article. The research presented in our paper is the first, preliminary study concerning the antiviral activity of Greek oregano, thyme and costmary EOs against CMV to have been conducted in Poland. The work suggested is a very good reference and we are going to include it in the more detailed studies that we plan to conduct in the future.
- The authors identify many components in essential oils. This is good. It is necessary to specify in more detail which of these components turn out to be the best antiviral substances? What exactly act on the cucumber mosaic virus? Obviously, different components of essential oils will act on different biochemical functions of different viruses.
Answer. I agree with this remark. However, this is the first, and thus a preliminary, Polish study into the antiviral activity of these 3 EOs against CMV. Further studies are going to be conducted in order to determine which particular EO components have the strongest antiviral activity.
- In this work, it is possible to indicate in more detail the relationship between the obtained data and those given in the literature. This will help you understand some of the principles and draw more solid conclusions.
Answer. In our paper, we have studied are antiviral activity of Greek oregano, thyme and costmary EOs against CMV. No data concerning the activity of these 3 EOs against CMV was found in the literature. Therefore, we believe that it is not possible for any more detailed observations to be made.
- 5. Unification of drawings is required.
Answer. Appropriate corrections have been made.
- It is necessary to describe in more detail the method of obtaining essential oils. If necessary, make a link to the methodology.
Answer. The essential oils were obtained according to the pharmacopoeial method, slightly modified. The reference was given.
- Conclusions look very modest. Please revise the conclusions substantially and expand them. The article presents many interesting results, on the basis of which important conclusions can be drawn.
Answer. Appropriate corrections have been made.
Round 2
Reviewer 1 Report
Dear Authors,
I congratulate you on your tremendous work.
However, I think that in the case of table 1, the header should contain precisely the parameters listed there (i.e., relative area (%)). In addition, you wrote in the text that:
"The percentage share of compounds identified in the EOs was computed by the normalization method from the GC peak areas."
So, the correct term would be Relative Area (%) or Relative Content (%) (because the respective values are relative, not absolute, and, of course, the unit of measure is not mm2).
Good luck and a happy new year!
Author Response
Review 1 (Round 2)
However, I think that in the case of table 1, the header should contain precisely the parameters listed there (i.e., relative area (%)). In addition, you wrote in the text that:
"The percentage share of compounds identified in the EOs was computed by the normalization method from the GC peak areas."
So, the correct term would be Relative Area (%) or Relative Content (%) (because the respective values are relative, not absolute, and, of course, the unit of measure is not mm2).
Answer. Appropriate corrections have been made.
